# *Kontinuous Kontext*: Continuous Strength Control for Instruction-based Image Editing

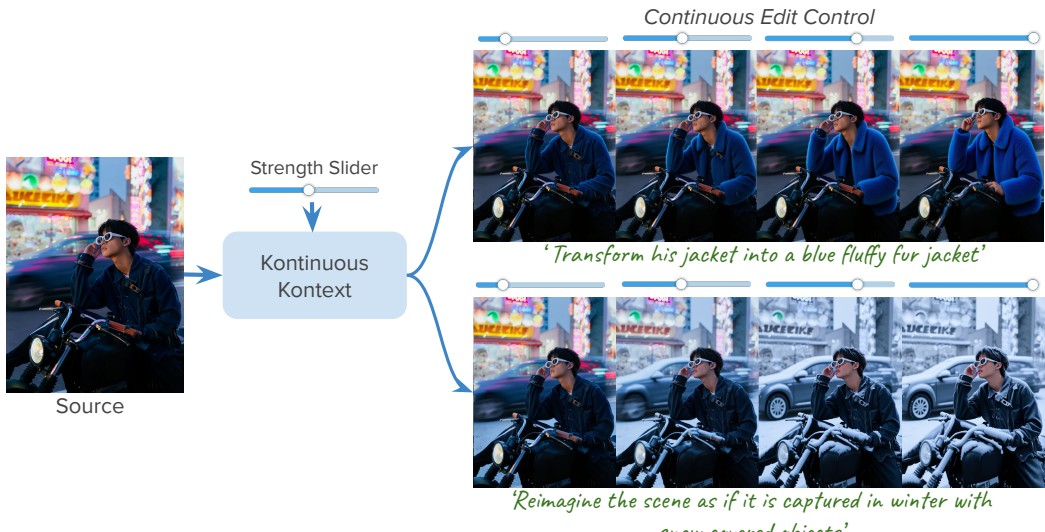

Figure 1: *Kontinuous Kontext* produces smooth edit trajectories across diverse attributes given an image, instruction, and an edit scalar strength. Unlike prior methods that require attribute-specific training, ours is a unified approach to enable fine-grained control.

## Abstract

Instruction-based image editing offers a powerful and intuitive way to manipulate images through natural language. Yet, relying solely on text instructions limits fine-grained control over the extent of edits. We introduce *Kontinuous Kontext*, an instruction-driven editing model that provides a new dimension of control over edit strength, enabling users to adjust edits gradually from no change to a fully realized result in a smooth and continuous manner. *Kontinuous Kontext* extends a state-of-the-art image editing model to accept an additional input, a scalar edit strength which is then paired with the edit instruction, enabling explicit control over the extent of the edit. To inject this scalar information, we train a lightweight projector network that maps the input scalar and the edit instruction to coefficients in the model's modulation space. For training our model, we synthesize a diverse dataset of image-edit-instruction-strength quadruplets using existing generative models, followed by a filtering stage to ensure quality and consistency. *Kontinuous Kontext* provides a unified approach for fine-grained control over edit strength for instruction driven editing from subtle to strong across diverse operations such as stylization, attribute, material, background, and shape changes, without requiring attribute-specific training.

## 1 Introduction

The advent of large-scale text-to-image generative models (Ho et al., 2020; Song et al., 2022; Rombach et al., 2022) has enabled phenomenal progress in instruction-driven image editing, allowing users to perform a broad range of edits through natural language instructions (Hertz et al., 2022;

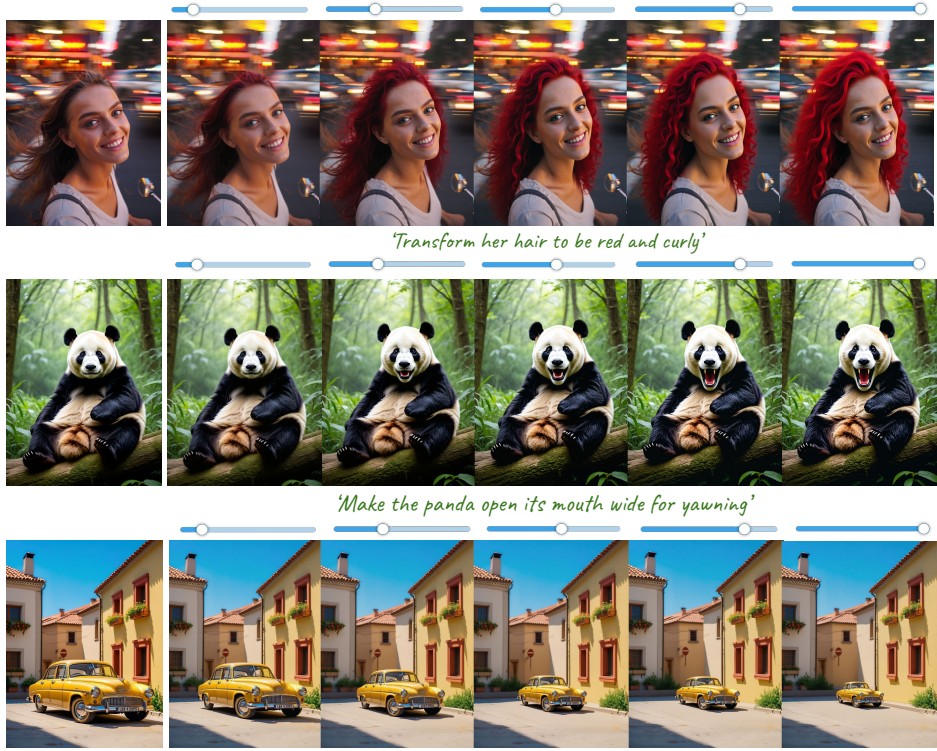

Figure 2: *Kontinuous Kontext* enables finer control across diverse edits. It can do simultaneous changes in attributes hair color and structure, highly localized changes such as editing the panda's mouth and geometric edits such as changing the size of the car.

Brooks et al., 2023; Batifol et al., 2025). With a single prompt (e.g., "make the person old"), these models can change style, modify object appearance or shape, and add or remove objects. While text is an intuitive interface for specifying editing goals, it is also a coarse modality: it conveys what change to make but not to what extent. As a result, users lack fine-grained control over the strength of an edit (e.g., adjusting the degree of "oldness" in a portrait). This limitation poses a central challenge for achieving precise and controllable image manipulation.

To address this challenge, prior work has explored continuous control for image manipulation, ranging from GAN-based latent space editing (Shen et al., 2020; Härkönen et al., 2020; Abdal et al., 2021; Patashnik et al., 2021) to diffusion-based methods that rely on specialized per-attribute modules (Cheng et al., 2025; Gandikota et al., 2024; Sharma et al., 2024). While these approaches demonstrate the appeal of continuous editing, they are often restricted to narrow domains or require dedicated training for each attribute. This leaves open the need for a unified method that enables continuous control across diverse types of edits without the burden of training per-attribute models.

In this work, we introduce *Kontinuous Kontext*, an instruction-driven image editing model that introduces a new dimension of control, enabling continuous adjustment of edit strength across diverse edit categories. Rather than being limited to a binary "before/after" operation, our approach enables smooth traversal between no edit and a fully realized edit, turning coarse instructions into rich, tunable controls. For example, users can gradually change the extent of stylization or intensity of snowfall (Fig. 1), as well as perform local edits with finer control including attribute edits such as hair color, facial expression, or object size (Fig. 2). By transforming discrete instructions into continuous editing trajectories, our method bridges the gap between intuitive text prompts and fine-grained user control, offering a level of precision unattainable with text alone.

We realize this new dimension of control by augmenting an existing instruction-based image editing model with an additional input scalar that specifies edit strength. Specifically, we build on Flux Kontext (Batifol et al., 2025), a state-of-the-art instruction-driven image editing model and condition it with the strength scalar via a lightweight projector network. The projector takes as input the scalar value together with the edit instruction embeddings and outputs coefficients calibrated to the specific

edit instruction. These coefficients operate in the model's modulation space (Garibi et al., 2025; Dalva et al., 2024), where they modulate the text tokens, effectively refining the edit instruction to reflect the desired strength.

Training the projector requires data consisting of source image, edit instruction, edit image, and annotations of edit strengths, which is not readily available for real images. To overcome this limitation, we synthesize such tuples using existing generative techniques. Specifically, we first use an LVLM (Bai et al., 2025) to generate diverse, image-specific edit instructions. Next, we apply Flux Kontext to produce edited images from the source images and the synthesized instructions. Finally, we use a diffusion based image morphing model (Cao et al., 2025) to generate intermediate edits at varying strengths. The synthesized data, however, often provides noisy supervision, where the sequences are not smooth or the intermediate images have artifacts or deviate too far from the endpoints. To address this, we apply filtering based on identity preservation of input images and smoothness of the edit transitions to obtain clean, reliable training data. In addition, the scale and diversity of the dataset helps mitigate remaining inaccuracies and outliers. Notably, we find that even when trained on this high quality filtered but moderately sized dataset, our method generalizes strongly across diverse editing categories.

Extensive experiments across a broad spectrum of instruction driven editing tasks show that *Kontinuous Kontext* provides rich, diverse, and finely controlled results. It enables precise strength control for local edits such as attribute, material or appearance changes, global transformations such as style or environment and lighting changes, and even challenging geometric edits like shape morphing. Notably, it generalizes beyond its training categories to unseen cases such as facial attribute and body shape changes. These findings establish our approach as a powerful, general framework for continuous instruction-driven image editing, opening new directions for fine-grained and controllable visual editing.

## 2 RELATED WORKS

**Instruction-driven Image Editing.** The advancements of scalable visual generative models (Esser et al., 2024; Podell et al., 2023; Ramesh et al., 2022; Wu et al., 2025a; Rombach et al., 2022) trained on internet-scale image-text pairs have fueled a wide range of image editing applications. Instruction-based image editing, introduced by Instruct-Pix2Pix (Brooks et al., 2023) enables editing images with text instructions. To this end, they curated a synthetic dataset of image-edit pairs generated using Prompt2Prompt (Hertz et al., 2022), with corresponding editing instructions generated by an LLM, and fine-tuned the Stable Diffusion model (Rombach et al., 2022) for instruction-driven editing. Subsequently, many works (Sheynin et al., 2024; Zhang et al., 2025; 2024b) have improved the dataset curation pipeline and model architecture, leading to stronger instruction-following ability. More recent approaches train large unified models for both generation and editing (Batifol et al., 2025; Wu et al., 2025a;b; Xiao et al., 2025). These models are capable of performing diverse editing tasks such as personalization, scene composition, and instruction-based editing. Despite their remarkable general-purpose editing capabilities, these models lack control over the extent of the edit, limiting their applicability for users who require fine-grained adjustments.

**Discovering Continuous Control in Generative Models.** A common approach to achieve control over edit strength is through traversals in latent spaces. In GANs and VAEs, compressed latent representations capture rich semantics, enabling the discovery of directions that correspond to semantic attributes (Karras et al., 2019; Härkönen et al., 2020; Hou et al., 2017; Higgins et al., 2017). Numerous traversal methods have been developed to leverage these directions for fine-grained attribute manipulation (Shen et al., 2020; Abdal et al., 2021; Patashnik et al., 2021). However, such methods remain restricted to narrow domains. Extending the idea of latent space traversal to diffusion models is challenging, as the denoising network does not naturally provide a compact latent space (Kwon et al., 2022), text embeddings are not smooth (Hertz et al., 2022), and LoRA-based weight interpolations (Gandikota et al., 2024; 2025; Dravid et al., 2024) remain computationally expensive and concept-specific. These approaches all rely on discovering latent or weight-space directions with continuous variation. In contrast, we augment the instruction mechanism with a new control dimension, enabling smooth adjustment of any attribute the model can already edit. Hence, our model does not require any additional training for specific attributes.

**Adding Continuous Control for Image Editing.** Another set of works introduces continuous control in diffusion models by either fine-tuning the model itself or training auxiliary encoders that modify its inputs. Some works (Sharma et al., 2024; Cheng et al., 2025; Magar et al., 2025) generate

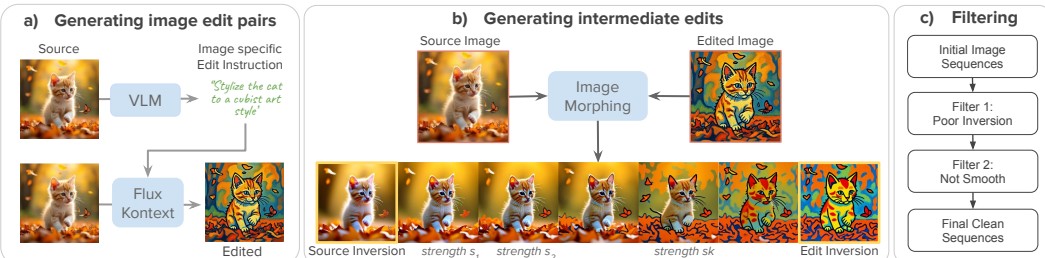

Figure 3: **Data generation.** Our pipeline consists of three steps: (a) We generate an edit instruction for each source image using a pretrained VLM, then apply Flux Kontext, an instruction-driven editing model, to produce a full-strength edit. (b) We synthesize intermediate-strength edits using a diffusion-based morphing method (Cao et al., 2025), which inverts both the source and edited images into the diffusion latent space and interpolates their features. (c) To compensate for inconsistencies in the morphing sequence (Fig. 5), we filter the samples based on the inversion quality and uniformity of the sequence.

synthetic data with varying material or illumination properties using rendering engines and fine-tune diffusion models for continuous control over these attributes. Others train encoders to predict new token embeddings injected into the text embedding space, enabling control over 3D properties such as orientation, illumination, and shadows (Cheng et al., 2024; Parihar et al., 2025; Burgess et al., 2024). A further line of work trains adapters that connect the continuous latent spaces of GANs with the stronger generative capabilities of diffusion models, specifically for face attribute editing (Parihar et al., 2024; Li et al., 2024). Despite their effectiveness, methods across these directions remain limited to a single attribute or object category.

**Image interpolation.** A promising baseline strategy to achieve continuous control in image editing could be to generate the edited image with instruction and then generate intermediate images between the source and the edited image. Diffusion-based morphing methods (Cao et al., 2025; Zhang et al., 2024a) aim to generate intermediate transitions by interpolating in the diffusion feature space, under the assumption that this space is semantically smooth. While this assumption holds in some cases, the space is not robust to outliers and often produces artifacts in intermediate morphs, such as missing objects or blurred scene content (Fig. 5). Another option is to adapt large video inbetweening models (Wan et al., 2025; Zhu et al., 2025; Wang et al.) to synthesize intermediate frames as continuous edits. However, as these models are trained on natural videos, they produce abrupt transitions for imaginative edits such as stylization or attribute changes, and their outputs frequently exhibit motion blur, making them unsuitable for high-quality image editing.

## 3 METHOD

We extend instruction-driven image editing by introducing a new dimension of control: continuous adjustment of edit strength. To this end, our approach has two key stages. First, we generate a diverse synthetic dataset of paired examples consisting of source images, edited images, edit instructions, and continuous strength values (Sec. 3.1). Second, we propose a simple yet effective approach: fine-tuning a modified instruction-driven editing model that accepts a scalar strength input alongside the edit instruction, enabling smooth and continuous control over the target edit (Sec. 3.2).

### 3.1 DATASET

Our method utilizes a dataset of tuples $(x, e, s, y_s)$, where $x$ is a source image, $e$ is an edit instruction, $s$ is an edit strength, and $y_s$ is the corresponding target edit. Since collecting real data with multiple strength levels is challenging, we curate a synthetic dataset using pretrained generative models. Our data generation process involves three steps: (i) generate a full-strength edit using an existing instruction-driven editing model, (ii) interpolate between the source and the full-strength edit to produce intermediate-strength variations, and (iii) filtering poor quality data samples.

**Generating Image Edit Pairs.** We begin by sampling $110K$ images of diverse objects and scenes across different background and environment conditions from the Subject200K dataset (Tan et al., 2024). For each image, we generate an edit instruction using Qwen LVLM (Bai et al., 2025), covering a diverse category of continuous editing operations (Fig. 3a). We categorize edits into global scene edits (*stylization*, *scene reimagination*, and *environment change*) and local object-specific edits (*material* and *appearance editing*, *attribute modification*, and *shape morphing*) also shown in

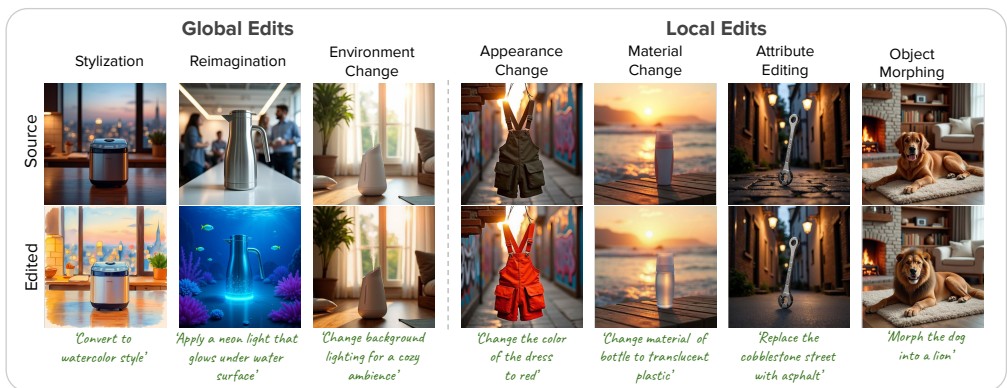

Figure 4: Samples from diverse image editing categories in our synthesized dataset. We cover a wide range of global edits, including stylization, reimagination, and environment changes, as well as local edits such as appearance changes, material changes, attribute editing, and object morphing.

Fig. 4. We define a fixed template system prompt for each subcategory. Additionally, we generate $n$ in-context examples using GPT4 for each of the subcategory. During instruction generation, we randomly sample from these category specific in-context examples to guide the VLM in generating diverse instructions. The source image and its corresponding instruction are then used to produce a full-strength edit ($y^*$) with Flux Kontext (Batifol et al., 2025). Generating the edit from Flux Kontext ensures consistency with the base model's output distribution. Further details of the prompts and additional samples are provided in the appendix Sec. A.2.

**Generating Edits With Intermediate Strength.**
We generate intermediate edits by synthesizing smooth transitions between the source image $x$ and the full-strength edit $y^*$ generated by Flux Kontext. We define a discrete set of $N+1$ edit strengths $\{s_i = i/N \mid i = 0, \dots, N\}$ uniformly sampled within the normalized range $[0,1]$. Here, $s_0 = 0$ corresponds to the unedited source, $s_N = 1$ corresponds to the full edit $y^*$, and the intermediate values $s_i$ for $1 \leq i \leq N-1$ represent proportionally graded changes. Given the source and edited images, we use off-the-shelf diffusion based image morphing method Freemorph (Cao et al., 2025) to generate the intermediate images $y_{s_i}$, which we treat as edits at the corresponding strengths $s_i$. Freemorph first inverts the two end point images into the latent space of pretrained diffusion model. Next, it performs guided spherical interpolation between their self-attention maps during denoising to produce intermediate morphs. This yields perceptually monotone transitions that interpolate between the two images (Fig.3b). We use prescribed $N = 6$ as provided in Freemorph (Cao et al., 2025).

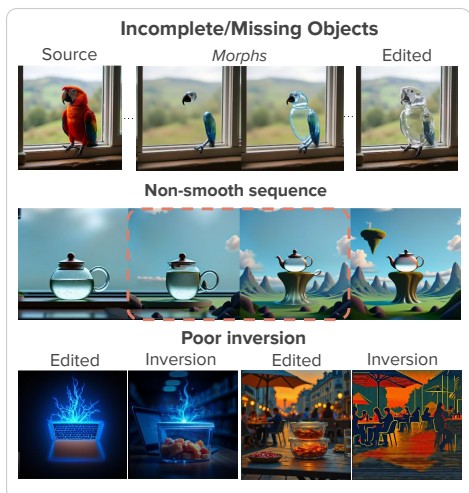

Figure 5: Generating intermediate images with Freemorph can introduce inconsistencies such as incomplete objects, abrupt jumps, or errors from diffusion inversion. We filter such cases to obtain a clean dataset with smooth trajectories.

We observe that Freemorph has two key limitations. First, its latent space is not semantically smooth, often producing unnatural intermediate images, artifacts with incomplete objects (Fig. 5) and abrupt transitions for large edit transformations. More broadly, as an inference-time heuristic, Freemorph lacks robustness, which further contributes to the errors. To address these issues, we employ an extensive data filtering pipeline. Second, since Freemorph relies on diffusion inversion, it introduces reconstruction errors in the source and edited images during inversion, which makes the intermediate images inconsistent (Fig. 3b). We fix this limitation by replacing the original endpoints with their reconstructions, ensuring consistency with the intermediate morphs.

**Data Filtering.** While effective, the above data generation pipeline is prone to errors from the underlying generative models (Fig. 5), making filtering essential to eliminate inconsistent samples. To filter out samples with non-smooth edit trajectories, we quantify the uniformity of the edit trajec-

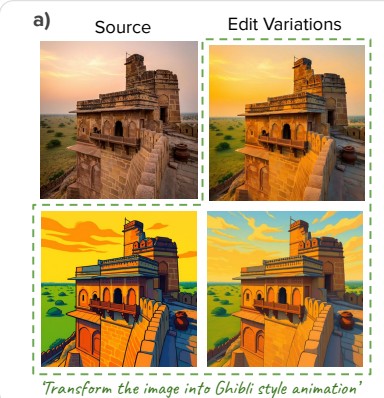 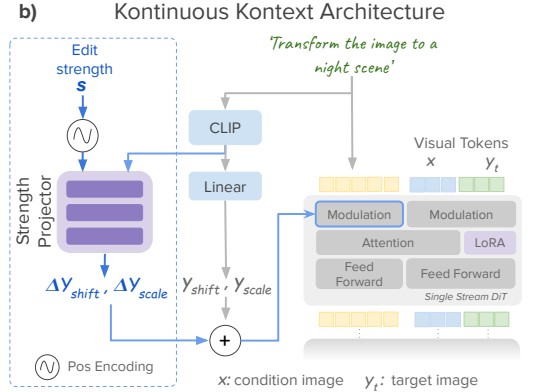

Figure 6: **Model architecture.** (a) In a simple experiment, we scale the text-token modulation parameters in Flux Kontext with a scalar to generate edit variations. This perturbation produces edits of varying strengths, revealing that modulation parameters can govern edit strength. (b) Building on this insight, we design a lightweight projector network that maps a scalar edit strength $s$ to offsets of the text modulation parameters, enabling precise control over edit strength.

tory and threshold on this score. For a training sample $(x, e, s, y_s)$, the extent of change between the source $x$ and edit $y_s$ should scale with the edit strength $s$. Equivalently, the distance between adjacent images in the sequence should remain consistent. We define the sequence of deltas as $D = \{d_{0,1}, d_{1,2}, \ldots, d_{N-1,N}\}$, where $d_{i,i+1}$ is the distance between image $y_i$ and $y_{i+1}$ and measure its uniformity via the KL-divergence from a discrete uniform distribution. Samples with divergence above $0.15$ are discarded.

In addition to non-uniform trajectories we observe for stronger edits, the diffusion inversion step in Freemorph can drastically alter the edited image (Fig. 5). We discard such cases by thresholding the image distance between the edit and its inversion. Similarly, in some cases Flux Kontext fails to perform the edit and instead reproduces the input with minimal changes; we filter out such examples by computing image distance between the source and edited images. We used LPIPS (Zhang et al., 2018) to compute the image distance in all the filtering criteria. After filtering, our dataset is reduced from $110,147$ to $64,613$ high-quality, smooth and, accurate edit trajectories. Additionally, we generate $10K$ object size change dataset by pasting objects in different sizes in black backgrounds.

## 3.2 KONTINUOUS KONTEXT

**Preliminaries.** We build our model on Flux Kontext (Batifol et al., 2025), a DiT-based instruction-driven image editing model. It takes a source image and an edit instruction as input and outputs the edited result. The design follows Flux (Labs, 2024), where image and text are encoded as tokens and processed through visual and textual attention streams. Flux Kontext extends this by encoding the source (context) image with the Flux autoencoder, then concatenating the source tokens ($x$) with the noised target tokens ($y_t$), which are jointly processed in the visual stream (Fig. 6). As in Flux, a pooled embedding of the edit instruction is fused with the timestep embedding to predict separate modulation parameters for both textual and visual tokens.

**Conditioning on edit strength.** Our goal is to inject the scalar edit strength into the instruction-driven Flux Kontext model (Batifol et al., 2025). Intuitively, edit strength can be viewed as an attribute of the instruction itself, which suggests representing it as an additional token in text token sequence. However, our early experiments revealed that the text embedding space is not a smooth latent space for strength control, often producing abrupt transitions between adjacent edit strengths (Fig. 15). Recent works (Garibi et al., 2025; Dalva et al., 2024) have shown that the modulation space of DiT models is highly disentangled and enables fine-grained control of attributes in text-to-image generation. In particular, object-specific attributes can be modified by adjusting the modulation parameters of the corresponding word in the text prompt (Garibi et al., 2025).

We find that the modulation space of instruction-driven image editing models allows control over edit strength. In a simple experiment, we scaled the modulation parameters of the text tokens with a scalar $v \in (0.5, 2.0)$ and generated multiple edits of the same image and instruction. As shown in Fig. 6 & appendix Fig. 14, perturbing the modulation parameters produce edits of varying strength, while preserving models prior of preserving image identity. Building on this insight, we inject edit-strength information into the network through the modulation parameters of the text tokens. Specifically, we design a strength projector network that maps the input scalar strength value to

offsets of the original text modulation parameters, enabling appropriate adjustments for continuous control of edit strength.

**Strength Projector** is a small MLP that maps the scalar edit strength $s \in (0, 1)$ into the offsets $[\Delta y_{shift}, \Delta y_{scale}]$ to the modulation parameters of the text tokens $[y_{shift}, y_{scale}]$. A direct implementation of this projector would predict identical offsets for all edits at a given strength, ignoring the type of edit. This leads to uncalibrated edits resulting in sudden jumps in edits. For example, as shown in Fig.7, for material editing, the model generates sudden transitions. To overcome this limitation, we provide

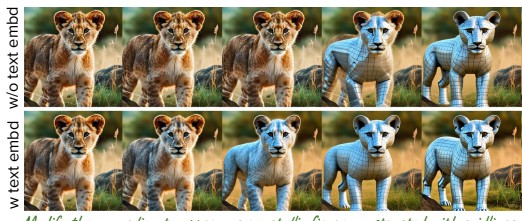

Figure 7: Adding text embeddings into the slider projector improves smoothness of edit transitions.

the pooled CLIP text embedding as an additional input, allowing the predicted modulation parameters to depend on the instruction. This results in calibrated modulations that enable smooth, continuous control across diverse edit categories. More details are in appendix Sec. A.3.

**Training.** We train our model on the curated dataset (Sec. 3.1) by sampling paired data consisting of source image $x$, edit instruction $e$, edit strength $s$, and target edit $y_s$. Trainable parameters include LoRA for the attention projection matrices of the Flux Kontext model, along with the projector network. Concretely, a data sample $(x, e, s, y_s)$ and a diffusion timestep $t$, we optimize the model using the standard flow matching loss:

$$\mathcal{L}_\theta = \mathbb{E}_{t \sim p(t), x, e, s, y_s} \left[ \left\| v_\theta(y_s^t, t, e, x, s) - (\epsilon - x) \right\|_2^2 \right], \tag{1}$$

where $y_s^t$ is the interpolated latent between $y_s$ and Gaussian noise $\epsilon \sim \mathcal{N}(0, 1)$, defined as $y_s^t = (1-t)y_s + t\epsilon$. $v_\theta$ is the Kontinuous Kontext model. As a regularization we randomly drop the slider conditioning with probability $0.1$. For more details are in Sec. A.1.

# 4 EXPERIMENTS

**Evaluation Benchmark.** We use a standard image editing benchmark, PIEbench (Ju et al., 2024), that consists of diverse and challenging instruction-driven image editing test examples. The benchmark consists of edits from the following editing categories: change object, add/remove object, change pose, change color, change material, change background and change style. We remove the add/remove category as it is not a continuous

| Methods | $\delta_{\text{smooth}} \downarrow$ | CLIP-Dir. $\uparrow$ |
|---|---|---|
| Diffmorpher | 0.371 | 0.181 |
| Freemorph | 0.365 | 0.189 |
| WAN-Video | 0.853 | **0.269** |
| Ours | **0.329** | 0.241 |

Table 1: Comparison with **Editing + Interpolation** baselines.

edit. The instructions involved challenging edits that often have two-three edits in one prompt (e.g., *'transform the dog into a brown german shepherd, while he stands on the bench'*). The evaluation dataset consist of $540$ images, with one edit instruction per image.

**Metrics.** We evaluate all the methods on two aspects: smoothness of edit trajectories and instruction following. Smoothness is measured with the triangle deficit ($\delta_{\text{smooth}}$), which captures second-order consistency between adjacent edits; smaller values indicate smoother transitions. We use DreamSim (Fu et al., 2023) as the distance metric and report the maximum deficit per sequence. A user study confirmed that this configuration for measur-

| Methods | $\delta_{\text{smooth}} \downarrow$ | CLIP-Dir. $\uparrow$ |
|---|---|---|
| ConceptSliders | 0.143 | 0.186 |
| Ours | **0.098** | **0.382** |
| MARBLE | 2.577 | **0.157** |
| Ours | **0.350** | 0.101 |

Table 2: **Domain specific** comparison.

ing smoothness of edits aligns best with human preference (Fig. 16). We evaluate the instruction following with CLIP directional similarity (CLIP-dir.) (Gal et al., 2021) aggregated over all edit strengths. Full details about metrics and evaluation for identity preservation are provided in Appendix A.6.

## 4.1 BASELINE COMPARISONS

We compare *Kontinuous Kontext* against two categories of baselines here, and with additional custom inference-based baselines in Sec. A.8:

**i) Editing + interpolation:** We first generate a full strength edit with Flux Kontext and then produce intermediate editing using interpolation methods. We use Diffmorpher (Zhang et al., 2024a),

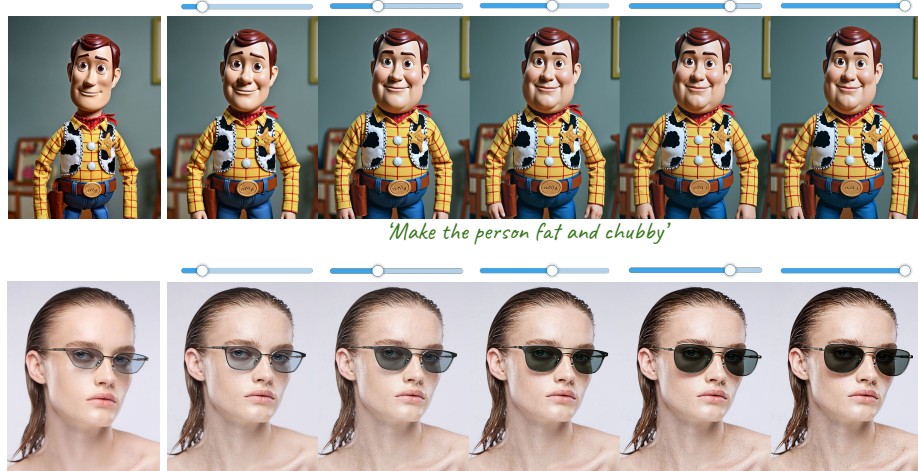

*'Make the person fat and chubby'*

*'Transform the glasses into aviator sunglasses'*

Figure 8: Our method enables continuous control for challenging geometric edits, including smooth body-shape transformations and seamless shape–color blending for eyeglass transition.

Freemorph (Cao et al., 2025), and a video inbetweening method WAN-2.1 (Wan et al., 2025) for interpolation and evaluate on PIEBench. Diffmorpher trains a LoRA on the two input images and interpolates the model weights, while Freemorph inverts the images and interpolates their attention features during denoising. Both are post-hoc heuristics applied to pretrained diffusion models, making them fragile across diverse edits. Video inbetweening methods, though explicitly trained for interpolation, perform poorly on imaginative stylization tasks since they are trained on real videos. Further, these baselines are slower as they require a cascade of models for slider based editing.

**ii) Domain specific methods:** Here, we compare against methods trained to control specific attributes, such as facial properties (e.g., age, smile) or material properties (e.g., transparency, metallicness). We compare with ConceptSliders (Kim & Ghadiyaram, 2025), which trains a LoRA module per attribute and achieves continuous control by weight interpolation. Because it is designed for continuous attribute control during image generation with diffusion models (and not for editing existing images), we evaluate it on 44 generated images across 11 sliders covering facial attributes, stylization, and scene edits. For material control, we compare with MARBLE (Cheng et al., 2025), which trains separate adapter networks to edit properties such as metallicness. We evaluate MARBLE on 40 PIEBench images from the material editing category on metallicness and glow properties.

**Analysis.** We present quantitative comparisons with interpolation methods in Tab. 1 and qualitative comparison in Fig. 9a on a challenging composite edit. Wan inbetweening abruptly transitions the color of the objects to the target full edit as such transformations are out of distribution for video model which is reflected as higher $\delta_{smooth}$ value. However, this also raises CLIP-dir., it does so only because the full edit appears prematurely at intermediate strengths.

| Methods | $\delta_{smooth} \downarrow$ | CLIP-dir $\uparrow$ |
|---|---|---|
| text-space condn | 1.468 | 0.191 |
| w/o text projector | 1.092 | 0.141 |
| w/o filtering | 0.483 | 0.228 |
| **Ours** | **0.329** | **0.241** |

Table 3: Ablation studies.

Diffmorpher and Freemorph introduce severe distortions in intermediate steps, often partially or completely removing the object, which leads to poor scores on both $\delta_{smooth}$ and CLIP-dir. Our method generates smooth transitions from the source to the final edit, gradually changing the color of the rock and ball while preserving their identity. We compare with domain-specific methods in Fig. 9b and Tab. 2. In comparison to ConceptSliders (C-Sliders), our method produces smoother transitions in appearance while preserving facial structure, as reflected in lower $\delta_{smooth}$. In contrast, C-Sliders often produces weak edits (see appendix for more comparisons), resulting in lower CLIP-dir. MARBLE, trained on synthetic 3D assets for material control, struggles on complex real images and, even when successful, exhibits abrupt jumps to the final edit at lower strengths. This leads to significantly higher $\delta_{smooth}$ despite high CLIP-dir. Our method achieves smooth and consistent transitions across diverse scenarios. Importantly, unlike domain-specific approaches that require attribute-specific training, our model works out of the box for new attributes, offering a single unified solution for continuous control of diverse attributes as shown in Fig. 8,& 11. We present additional comparisons in appendix Fig. 19, 20, 21& 22.

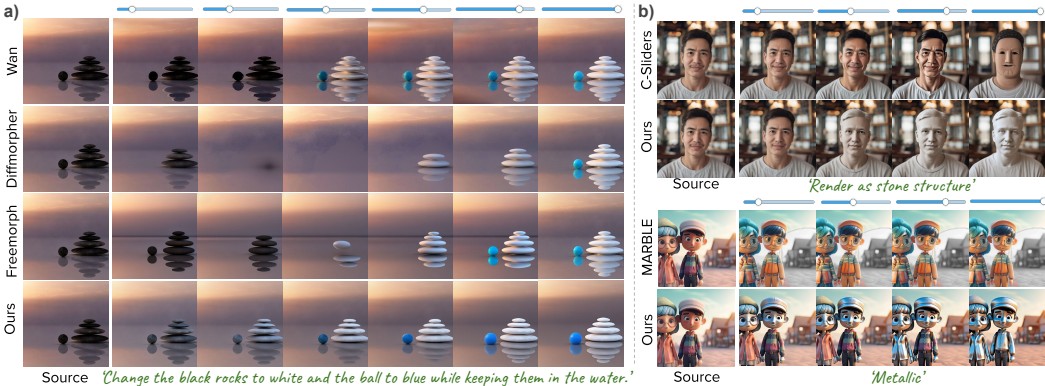

Figure 9: **Visual Comparison.** We evaluate against (a) image interpolation methods, where we first generate a full strength edit with Flux-Kontext and interpolate to obtain intermediate edits, and (b) domain-specific methods, which train separate LoRAs/Adapters for each attribute. Our generalized method achieves superior slider control with consistent image identity and smooth edit transitions.

### 4.2 ABLATIONS

We ablate design choices in Tab. 3. Conditioning by adding the slider projector output as an extra text token (**text-space condn**) is ineffective for fine-grained strength control and produces abrupt transitions, reflected in the worst $\delta_{smooth}$. Removing the pooled text embedding input from the slider projector (**w/o text projector**) leads to weaker, non-smooth edits and inferior $\delta_{smooth}$ and CLIP-dir. scores (see Fig. 15). Finally, effective data filtering that removes poor-quality and non smooth edit sequences from the dataset significantly improves both smoothness and text alignment.

### 4.3 USER STUDY

We conducted a user study to subjectively evaluate our method against all baselines. The study followed a head-to-head comparison where for each trial, one baseline was randomly selected, and its outputs were compared with ours across four dimensions: smoothness of the edit sequence, realism of the edits, editing capability with respect to the given instruction, and overall sequence quality. For each baseline, we sampled 20 input images, resulting in a total of 100 images evaluated. The study involved 20 participants, each providing judgments on the paired outputs. Figure 10 reports the win rates

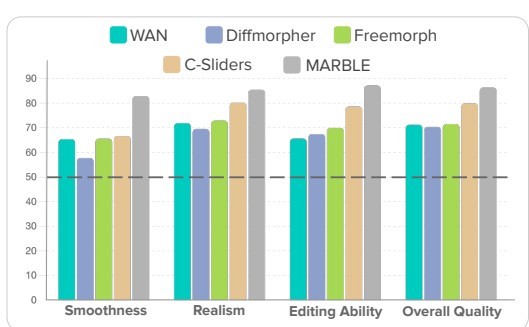

Figure 10: User study win-rates (%) of our method against baselines in pairwise comparisons.

of our method over the baselines. Morphing-based methods often appear smooth due to continuous transitions but suffer from artifacts or missed edits. Our method consistently outperforms all baselines across all criteria, delivering both faithful edits and superior perceptual quality.

## 5 DISCUSSION AND CONCLUSIONS

We presented *Kontinuous Kontext*, a simple extension to Flux Kontext that adds a continuous control dimension for instruction-driven image editing. Our method provides smooth, fine-grained control over the intensity of edits, without sacrificing the strong baseline capabilities of the underlying model. While highly effective for continuous edits, our approach has some limitations. For inherently discrete transformations, such as inserting or removing objects, the transitions are necessarily abrupt since there is no natural continuum. Moreover, as *Kontinuous Kontext* is built on Flux Kontext, it inherits its weaknesses in categories like precise geometric manipulations such as accurate object rotation or translation, where the base model itself struggles. A failure case of our method is in generating consistent extrapolating edits (Fig. 24) for large transformations.

Beyond its practical utility, this work highlights that edit intensity is naturally encoded in the modulation space of instruction driven diffusion models. By learning a lightweight projector into this space, we unlock a flexible control mechanism that generalizes across diverse edits without attribute specific training. This suggests that other forms of continuous control, such as spatial or temporal intensity fields, may be introduced in a similarly lightweight manner, opening opportunities for interactive editing tools that combine the richness of language with the precision of continuous sliders.

**Reproducibility statement:** We will release the code, pretrained models, and both the filtered and raw datasets used in this project. Our model is built on the open-source FLUX.1-Kontext dev image editing model. Details of the training setup and compute requirements are provided in Sec. A.1. A full explanation of dataset generation and filtering, along with representative examples, is given in Sec. 3.1 and Sec. A.2. The evaluation datasets and metrics are described in Sec. 4 and Sec. A.6. All baseline methods were evaluated using their publicly available code.

**Ethics Statement:** Our work focuses on continuous strength control for image editing, improving the controllability of image manipulation. While such techniques could be misused for creating deceptive or harmful content, similar to other generative models, outputs from our method can be watermarked. Our contributions are intended for research in controllable image generation, and we see this as enabling many positive applications. In particular, our approach can support creative design, accessibility, and educational tools, while ongoing advances in detecting AI-edited images further help mitigate risks of misuse.

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

# A APPENDIX

## A.1 IMPLEMENTATION DETAILS.

We train slider projector along with a rank-4 LoRA on all attention layers of the base diffusion model. We train all our models at a resolution of $512X512$. After filtering our dataset consists of $66K$ edit trajectories, along with their edit instructions. We train all models on a single NVIDIA A100 (80GB) GPU for $110,000$ iterations, using an effective batch size of $8$ and a constant learning rate of $2 \times 10^{-5}$. Training takes about $120$ hours to complete. During training, we drop the slider conditioning $10\%$ of the time. For inference, we use the default Euler scheduler from Flux Kontext and use $T = 28$ inference steps for generation. The generation time is similar to Flux Kontext model, as we only have the projector as the new component.

## A.2 DATASET GENERATION

In this section we provide the details about our dataset generation process:

**Generating Image Edit Pairs.** We use Subject200K (Tan et al., 2024) dataset to source our input images. This dataset has a diverse variety of input object and scenes captured in different environment conditions. We extract $110K$ source images from this dataset. Next, we generate image specific edit instructions for source images using a Qwen-VLM (Bai et al., 2025). For a good diversity of our dataset, we categorize our edit categories into global edits (stylization, scene reimagination and environment change) and local edits (material and appearance editing, attribute modification and shape morphing). For each image in the dataset, we randomly sample one of these editing categories, and ask VLM to generate instruction from that category. We pass the input image along with the system prompt to the multimodal VLM to generate instructions specific to the image. We use the following system prompt and ask the VLM to generate the edit instruction in a desired .json format for 'appearance change' edit, and use similar system prompts for other editing categories.

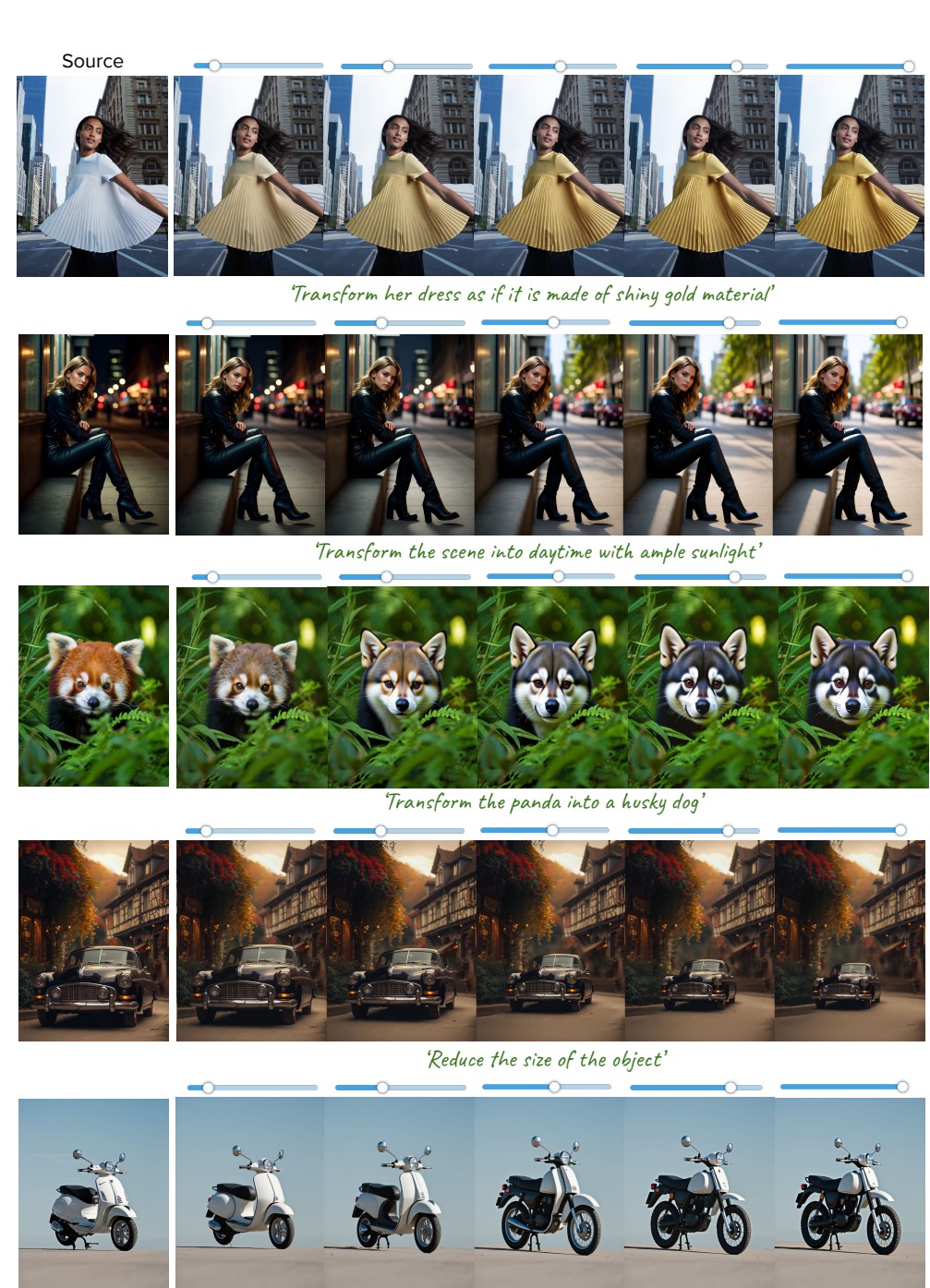

Figure 11: *Kontinuous Kontext* can enable fine-grained control over the edit strength for diverse instruction-driven image editing operations.

**System prompt for generating edit instructions**

System Prompt: You are a professional image editor. Generate an original, diverse, and detailed local appearance change instruction for the given object in the image. Create a unique instruction different in wording and content from the examples. Examples: {examples}
Output ONLY a valid JSON object with EXACT keys "category" and "instruction". No additional text or explanation.
Example output: {"category": "Appearance_Change", "instruction": "Modify the fabric of the couch to a rich burgundy velvet with gentle sheen."} DO NOT include trailing commas or escape characters.

We sample a predefine set $50 - 100$ in-context examples per edit category and randomly sample $4$ examples and combine it with the system prompt to generate rich prompts for generating diverse editing instructions. Here are the in-context examples for each of the categories in our dataset.

**In context example for local edits fonttitle**

**Appearance_change**
examples = [ "Transform the chair into plush candy-colored marshmallow material with soft reflections",
"Make the bicycle frame appear as flowing liquid metal with dynamic highlights",
"Turn the lampshade into glowing crystalline material with internal refracted light"]

**Material_change**
examples = [ "Replace the chair's wooden legs with polished chrome metal, emphasizing its reflective specularity",
"Make the tabletop appear carved from dark mahogany wood with visible grain and a semi-matte roughness",
"Transform the bag's fabric into smooth black leather with glossy highlights and subtle texture"]

**Attribute_change**
examples = ["Open the laptop lid halfway to reveal the keyboard",
"Rotate the handlebar of the bicycle by 45 degrees",
"Raise the adjustable lamp arm to maximum height"]

**Intra_object_morph**
examples = ["Morph a teapot into a lantern while keeping the spout as a decorative handle",
"Transform a bicycle into a motorbike with parts composed naturally",
"Morph a chair into a bench while preserving the backrest shape"]

---

**In context example for global edits**

**Stylization**
```
examples = [ "Render the scene in Studio Ghibli style with
dreamy backgrounds and soft pastel hues",
"Transform the image into Pixar-style 3D animation with vibrant
colors and cinematic lighting",
"Stylize the composition as a Van Gogh oil painting with thick
impasto brush strokes"]
```

**Environment_change**
```
examples = ["Blanket the entire landscape with fresh, thick
snow, covering trees and rooftops with crystalline frost",
"Transform the scene into a harsh winter blizzard with swirling
snow and reduced visibility",
"Age the entire scene to look like a weathered medieval village
with cracked stone walls"]
```

**Scene_reimagination**
```
examples = ["Place the entire village on a massive turtle's
back slowly moving through the ocean",
"Transform the bustling marketplace into a floating bazaar
carried by hot air balloons",
"Reimagine the city skyline as colossal crystal formations
reflecting rainbow light"]
```

**Generating image edits.** We use the source images and obtained editing instructions to generate edited versions of the source image using Flux Kontext (Batifol et al., 2025). Flux-kontext being a generalist editing model, it can generate high quality edits for the source images. However, in some cases it does not perform the edit and outputs the same input image. We filter our such cases in our filtering stage discussed next. Next, we present a qualitative subset of source, edit image and the instructions used for generating those edit images in Fig. 12.

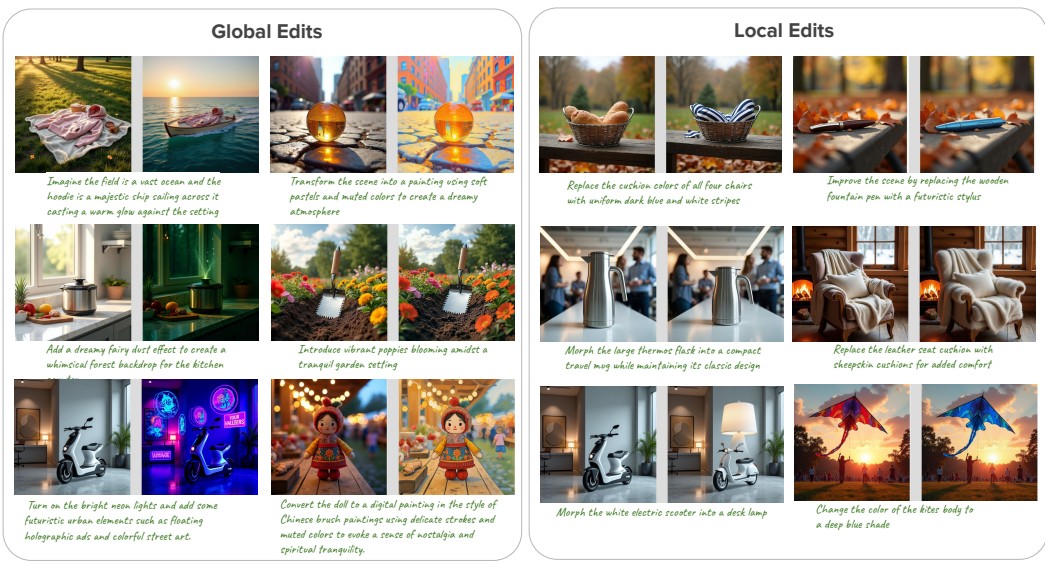

Figure 12: Samples for generated edit instructions and the generated edits from Flux Kontext

**Generating intermediate edits with Image morphing (Cao et al., 2025)** Given the source and edited image, we use Freemorph - a training free Diffusion based image morphing approach.

Freemorph requires input caption for the two source images to be interpolated. To this end, we use LLaVA (Liu et al., 2023) model to generate captions, as they suggested in their paper. Further, the method first inverts the two images and then interpolated the attention features during denoising. This requires a full denoising process to generate one morph image. In practice, we generate $N = 5$ intermediate morphs between the source and the edited image. We use official code provided by the authors that is built on StableDiffusion-2.1 Rombach et al. (2022) and use DDIM scheduler for generation with $T = 50$ steps. All the interpolations were generated at a native resolution $768X768$ of SD-2.1.

**Data Filtering.** We filter out the edit sequences that are not smooth and have significant inversion during the diffusion inversion. We visualize some examples that are selected and filtered out based on the reconstruction quality and sequence uniformity in Fig. 13.

## A.3 MODEL ARCHITECTURE

Our projector is a 4-layer MLP with dimensions $1536 \rightarrow 256 \rightarrow 128 \rightarrow 6192$. The output dimension of $D = 6192$ is divided into two chunks each of 3096 represnting offsets for modualtion parameters - $\Delta y_{scale}$ and $\Delta y_{shift}$. The 1536 dimensional input to the model consists of embedded scale value $s$ of dimension 768 and pooled CLIP text embedding of dimensions 768. We first apply sinusoidal positional encoding to $s$ to bring it to 128 dimensions followed by a linear layer to transform it to similar dimension of 768. The CLIP embedding and the encoded scale embeddings are concatenated and passed as a single input to the projector network.

## A.4 INFERENCE-TIME CONTROL IN MODULATION SPACE

We performed a simple experiment to analyse the effect of modulation-parameters on the edit images. We scale the modulation parameters with $v = (0.5, 1.3)$ for the text token and visualize the generated edit image in Fig. 14. Though the generated edits are diverse for different scale values, the scaling value $v$ does not directly correlate with the strength of the edit. This raises a need of learning a calibrated mapper like our slider projector, that can expose the strength control by accurately manipulating the modulation parameters.

## A.5 ABLATION STUDY

We present ablation study in Fig. 15 for different architecture choices. Adding the output of slider projector in the text embedding space leads to edit transitions with abrupt jumps. Similarly, adding without adding the pooled text embedding in the projector leads to non-smooth edit trajectory. Our design of injecting the slider control in the modulation space and making the projector adapt to the edit instruction embedding, results in smooth trajectories, enabling fine-grained control to the user.

## A.6 EVALUATION METRICS

### A.6.1 SMOOTHNESS OF THE EDIT SEQUENCE

We measure both first and second-order smoothness of an edit trajectory for quantitative evaluation. For a given source image $x$ and edit instruction, we generate a sequence of $N$ edited images $\{y_{s_1}, y_{s_2}, \ldots, y_{s_N}\}$, and include the source image as the initial element $y_{s_0} = x$, yielding a sequence of $N+1$ images. We use an image distance metric $d(\cdot, \cdot)$ to compare the images. We used Dreamsim (Fu et al., 2023) as it better captures the semantic differences between images in contrast to LPIPS that has a high spatial bias.

**First-order smoothness.** We define adjacent distances between the images in the sequence as

$$d_i = d(y_{s_i}, y_{s_{i+1}}), \quad i = 0, \ldots, N-1,$$

and compute the cumulative path length

$$L = \sum_{i=0}^{N-1} d_i.$$

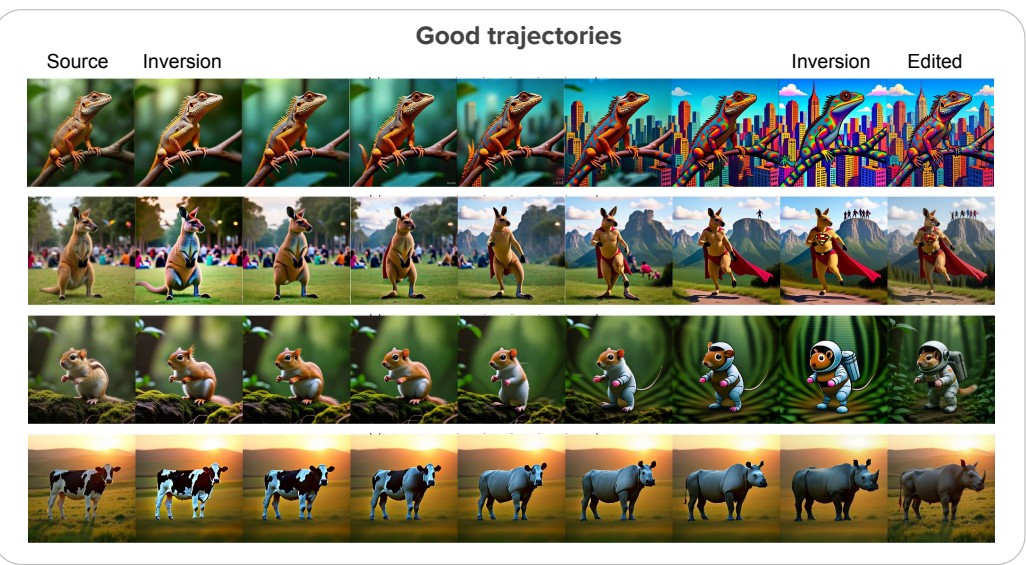

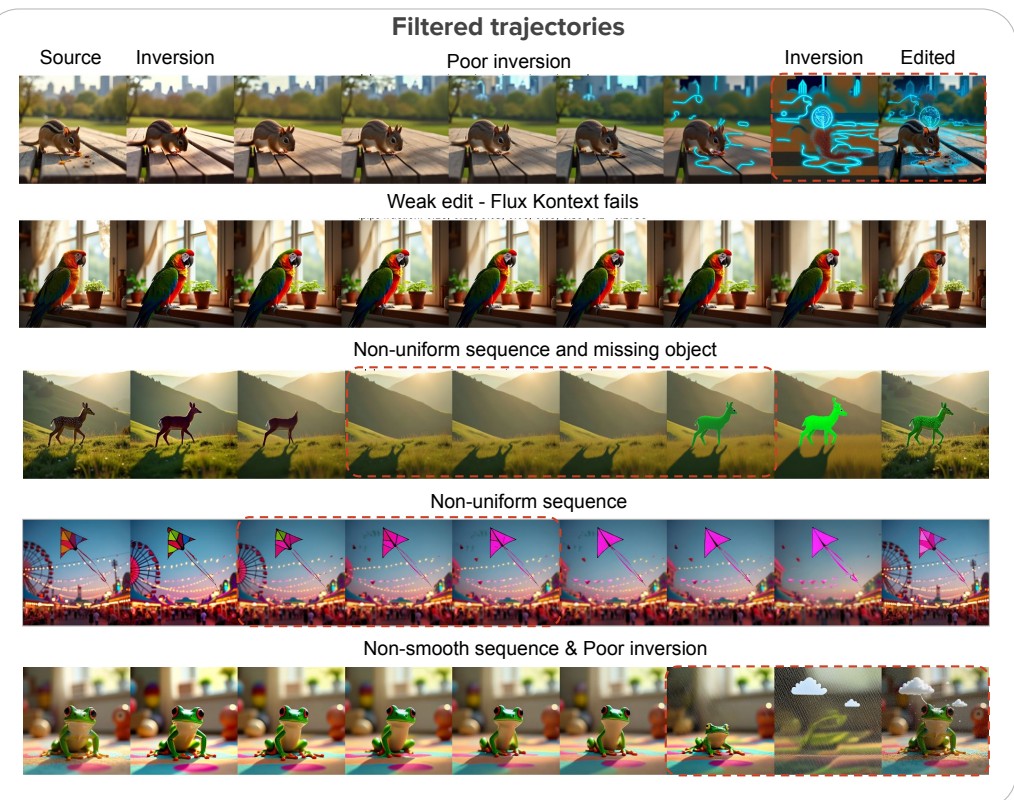

Figure 13: Samples trajectories from our synthesized dataset

**Scaling the modulation parameters at inference time**

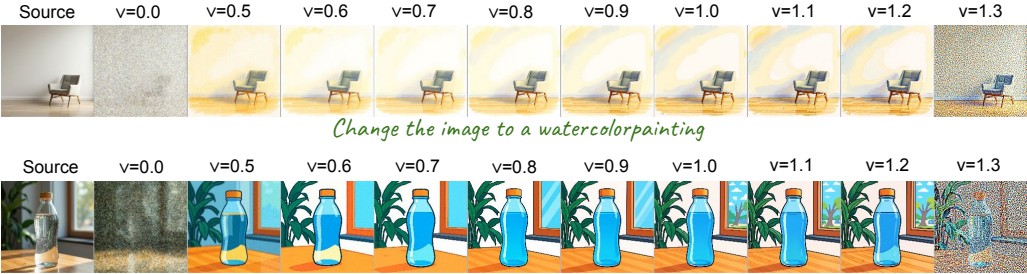

*Change the image to a watercolorpainting*

*Change the image to a cartoon*

Figure 14: **Inference time control in modulation space.** We conducted a simple experiment by scaling the text modulation parameters with values of $v \in (0.5, 1.3)$ to generate multiple edits. While these edits varied across different scales, the variations did not consistently correlate with the intended edit strength. This highlights the need for a dedicated learning module that can translate such variations into user-interpretable strength control by accurately manipulating the modulation parameters.

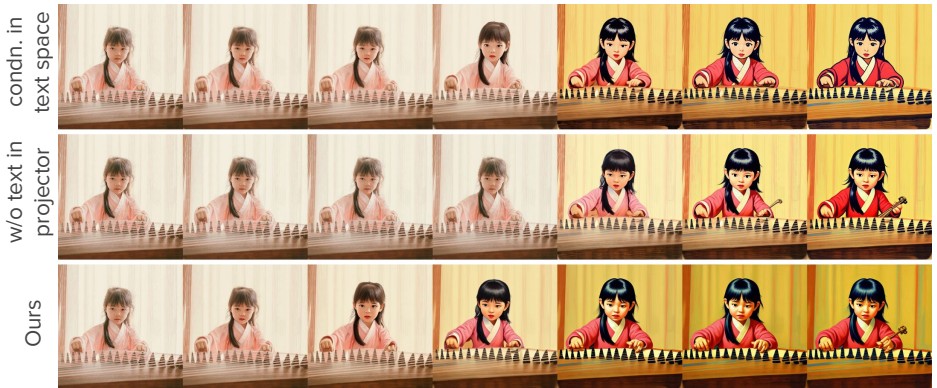

*Render the scene as an oil painting featuring a cartoon-style little girl playing a musical instrument.*

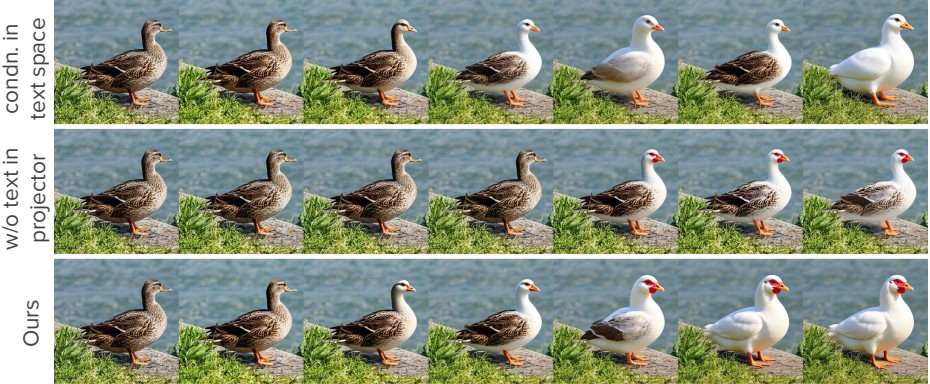

*Change the duck into a chicken sitting on a board near the water.*

Figure 15: Ablation over architecture of *Kontinuous Kontext*.

The first-order smoothness is then computed as:

$$\delta^1 = \max_i \frac{d_i}{L},$$

which captures the largest normalized jump in the trajectory.

**Second-order smoothness.** For local consistency, we compute the triangle deficit given by

$$\Delta_i = d(y_{s_i}, y_{s_{i+1}}) + d(y_{s_{i+1}}, y_{s_{i+2}}) - d(y_{s_i}, y_{s_{i+2}}), \quad i = 0, \ldots, N-2.$$

Each deficit is normalized by the direct distance between the endpoints:

$$\tilde{\Delta}_i = \frac{\Delta_i}{d(y_{s_i}, y_{s_{i+2}})}.$$

The second-order smoothness is then computed as:

$$\delta^2 = \max_i \tilde{\Delta}_i,$$

where smaller values indicate smoother local transitions.

**Analysis.** We conducted a user study to evaluate how well smoothness metrics align with human preferences. Participants were shown pairs of edit sequences and asked which appeared smoother in terms of transitions. The study included 20 volunteers and 40 sequence pairs. For each sequence, we computed first- and second-order smoothness using two distance functions: LPIPS (Zhang et al., 2018) and DreamSim (Fu et al., 2023). We then measured agreement between user choices and each of the four metric configurations (Fig. 16). Results show that $\delta^2$ (DreamSim) aligns best with user preferences, as it captures fine-grained semantic changes reflected in slider adjustments. While first-order smoothness prevents abrupt jumps, second-order smoothness ensures consistency in the rate of change, producing natural and continuous transitions that match user expectations. Fig. 17 illustrates this: although Sequence 1 has better first-order smoothness (lower $\delta^1$), Sequence 2 is semantically smoother, captured by a lower $\delta^2_{\text{smooth}}$. From these findings, we define the smoothness metric as:

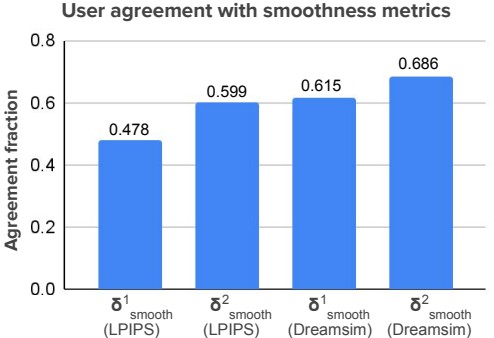

Figure 16: We performed one user study where we compute the alignment of the users scores given for smoothness of the sequence with the different variations of smoothness metrics. We found $\delta^{(2)}_{smooth}$ aligns well with the user preferences for smoothness indicating that it is a good metric to measure the smoothness.

$$\delta_{smooth} = \delta^2(Dreamsim)$$

### A.6.2 INSTRUCTION FOLLOWING WITH CLIP DIRECTIONAL SIMILARITY

For a given input image $x$, and edit instruction $e$, we edit the image with uniformly sampled edit strengths $\{s_i = i/N | i = 1, ..., N\}$ to obtain the edited image sequence $\{y_i | i = 1, ..., N\}$. We compute the CLIP-direction similarity for each of the edits at each strength as:

$$d_i = d_{clip-sim}(y_{s_i}, x, e), \quad i = 1, ..., N$$

and report the aggregated normalized CLIP-sim as:

$$D_{clip-dir} = \frac{\sum_{i=0}^{N}(d_i/s_i)}{N}$$

adjusting the directional similarity based on the edit strength.

### A.6.3 IMAGE IDENTITY PRESERVATION WITH CLIP IMAGE SIMILARITY

We quantify the image identity preservation by computing the CLIP-Image similarity between the source image and the edited image across different edit strengths. We present plot of the image similarity value across the edit strengths in Fig. 18. Our method, gradually reduces the image similarity with increasing strength following almost a linear decay. This further supports our finding that our method generates smooth transitions between subsequent images.

Edit sequence 1

$\delta^1_{smooth} = 0.350$     $\delta^2_{smooth} = 0.185$

Source

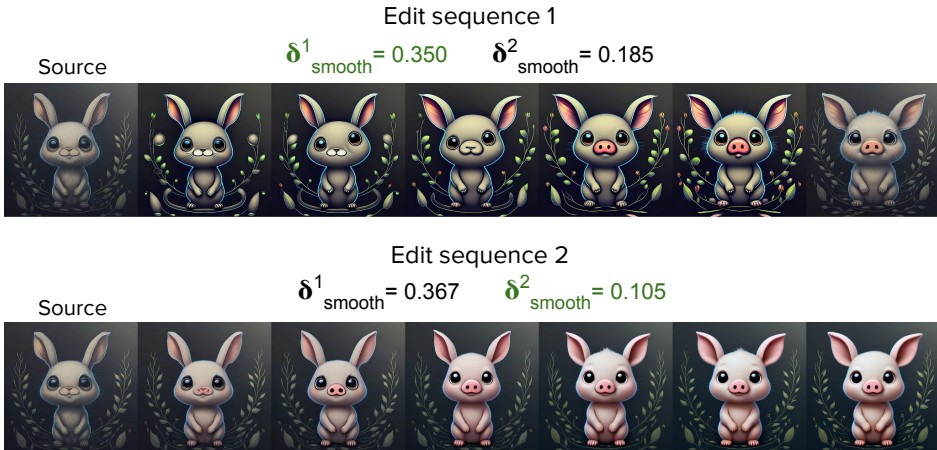

Edit sequence 2

$\delta^1_{smooth} = 0.367$     $\delta^2_{smooth} = 0.105$

Source

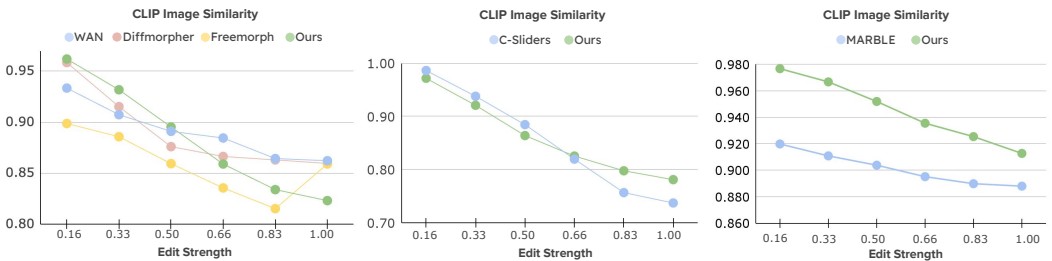

Figure 17: **Qualitative interpretation for first order and second order smoothness.** For slider-based image editing, second-order smoothness is more important than first-order smoothness, as it captures the local consistency needed for gradual, nuanced changes with slider controls.

Figure 18: **Comparison for identity preservation of our method against baselines.** Our method smoothly transforms the image into target edit over different edit strengths, resulting in close to linear decay in identity change and preserving identity well in lower strengths. In contrast, baselines change the identity of the subject significantly even with small edit strengths and don't change the image for stronger edits.

## A.7    QUALITATIVE COMPARISON

We present additional comparison results with interpolation based baselines in Fig. 19, 20 and with domain specific method ConceptSliders in Fig. 22, MARBLE in Fig. 21.

## A.8    ADDITIONAL BASELINES

We compared *Kontinuous Kontext* with two additional simple baselines: a) CFG-Scale - We change the classifier free guidance scale to control the extent of the edit, as we expect with higher cfg scale the generated edit should follow the edit instruction more closely. b) Attention reweighting - We scale the cross-attention maps between the text tokens and the generated visual tokens inspired by Prompt2Prompt (Hertz et al., 2022).

| Methods | $\delta_{smooth} \downarrow$ | CLIP-dir $\uparrow$ |
|---|---|---|
| CFG-scale | 152.205 | **0.242** |
| Attention-weighing | 120.760 | 0.237 |
| Ours | **0.329** | 0.241 |

Table 4: Experiments for comparison with additional inference time baselines.

The insight is that, if we increase the cross-attention weight with the text instruction the edited image will pay more attention to the edit resulting in stronger edits. We present comparison in Tab. 4 and Fig. 23. Both the methods fail the generate smooth edit transitions and distort the input image

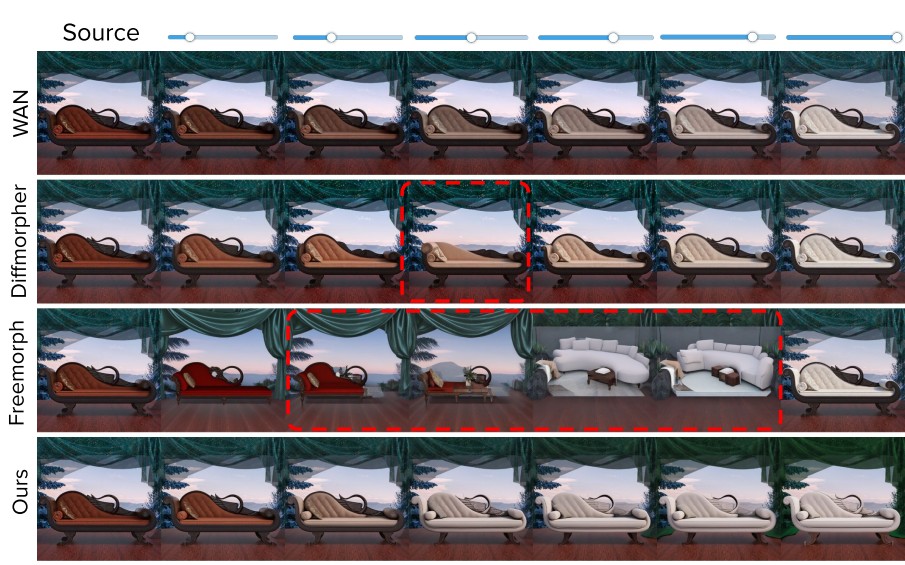

*Modify the couch to have a cottony texture and change the curtain to a green wool fabric.*

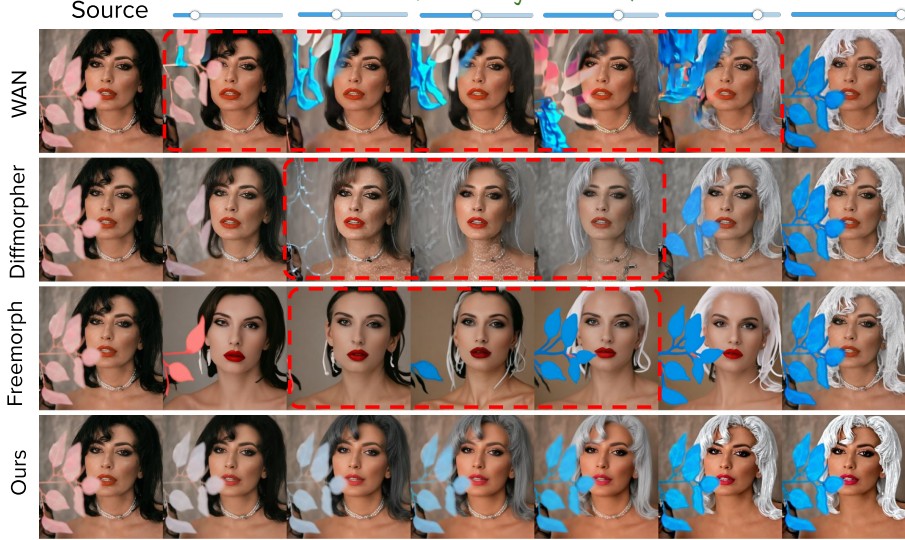

*'Modify the woman's hair to silver and change the flower to blue.'*

Figure 19: **Comparison with interpolation baselines.** Morphing-based methods generate smooth transitions; however, they often introduce artifacts in the intermediate images or omit details such as leaves. Similarly, the video inbetweening model WAN produces strong artifacts in intermediate frames, as these appearance transitions are out of domain for an inbetweening model trained only on real data.

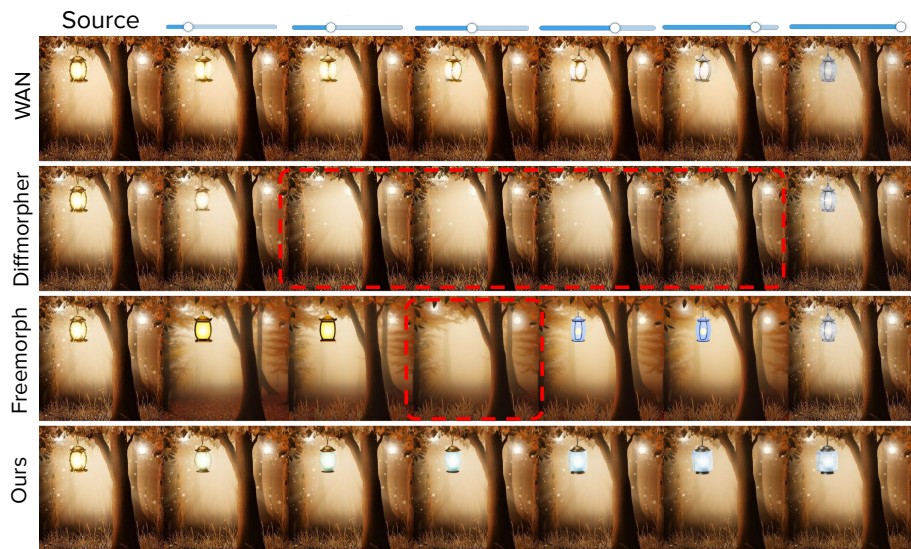

*'Modify the lantern to be crystal and the trees to appear ancient while retaining the forest setting with lights.'*

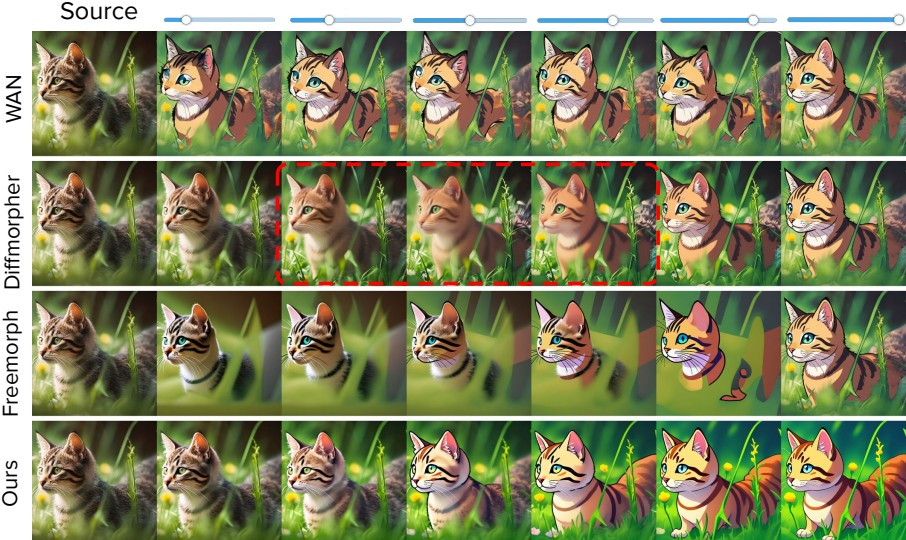

*'Transform the cat into an anime style in a digital art format'*

Figure 20: **Comparison with interpolation baselines.** DiffMorpher and FreeMorph remove objects in the intermediate edits of the first examples. Moreover, DiffMorpher produces blurred outputs even for simple stylization transitions. The WAN inbetweening model generates transitions with abrupt jumps in both examples. In contrast, our method produces smooth transitions while preserving image identity.

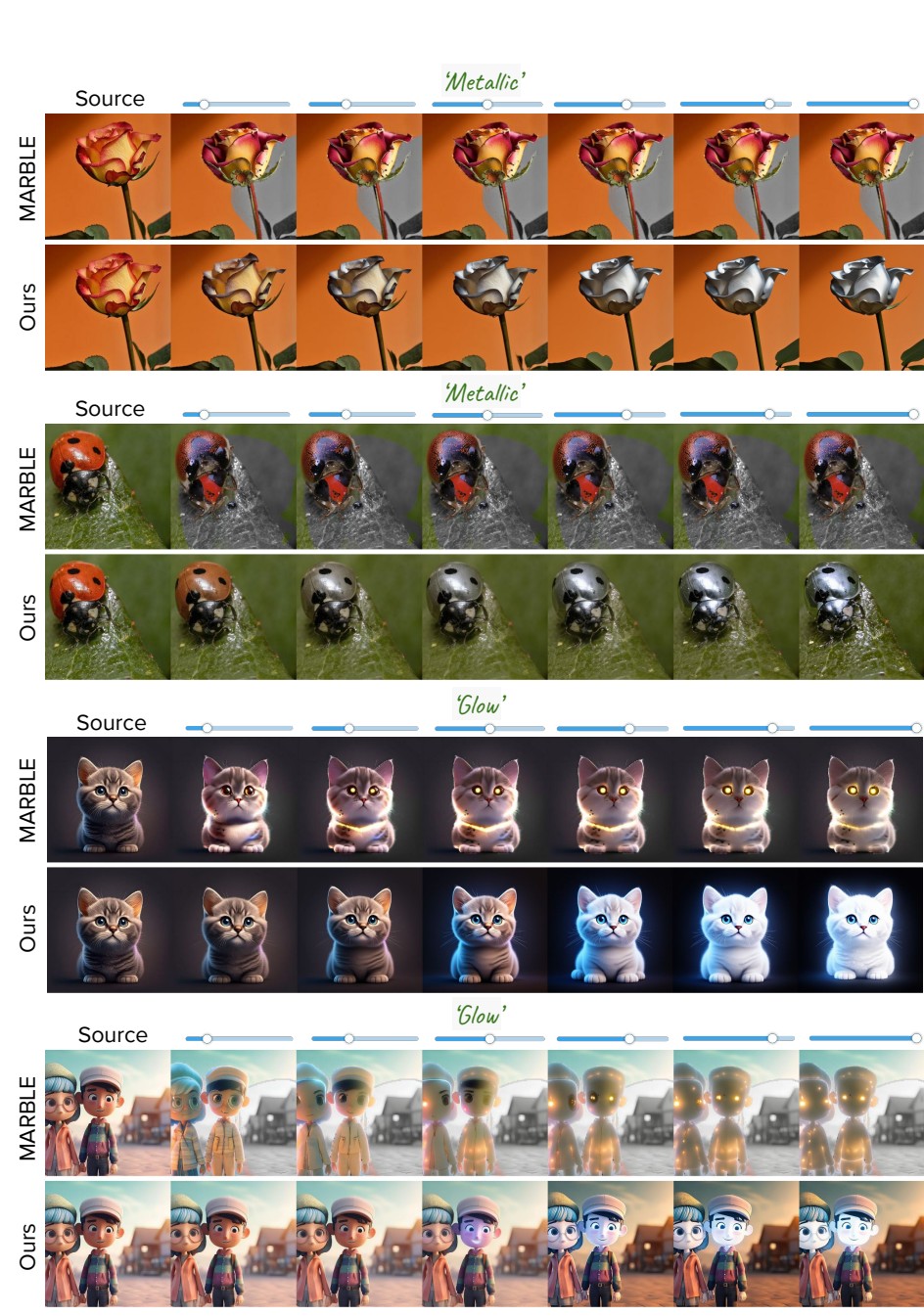

Figure 21: Comparison with MARBLE for material control

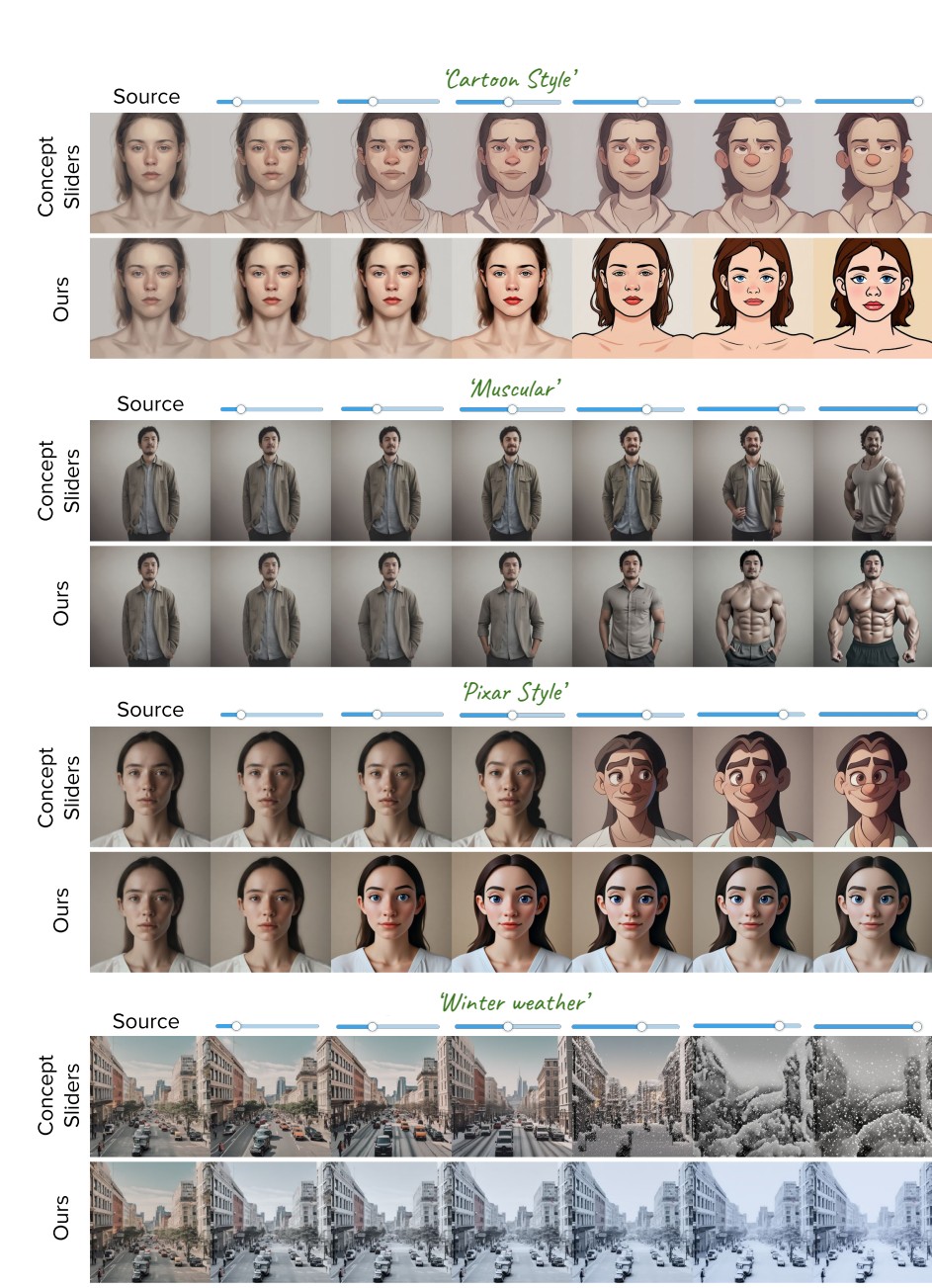

Figure 22: Comparison with Concept Sliders for diverse attribute editing.

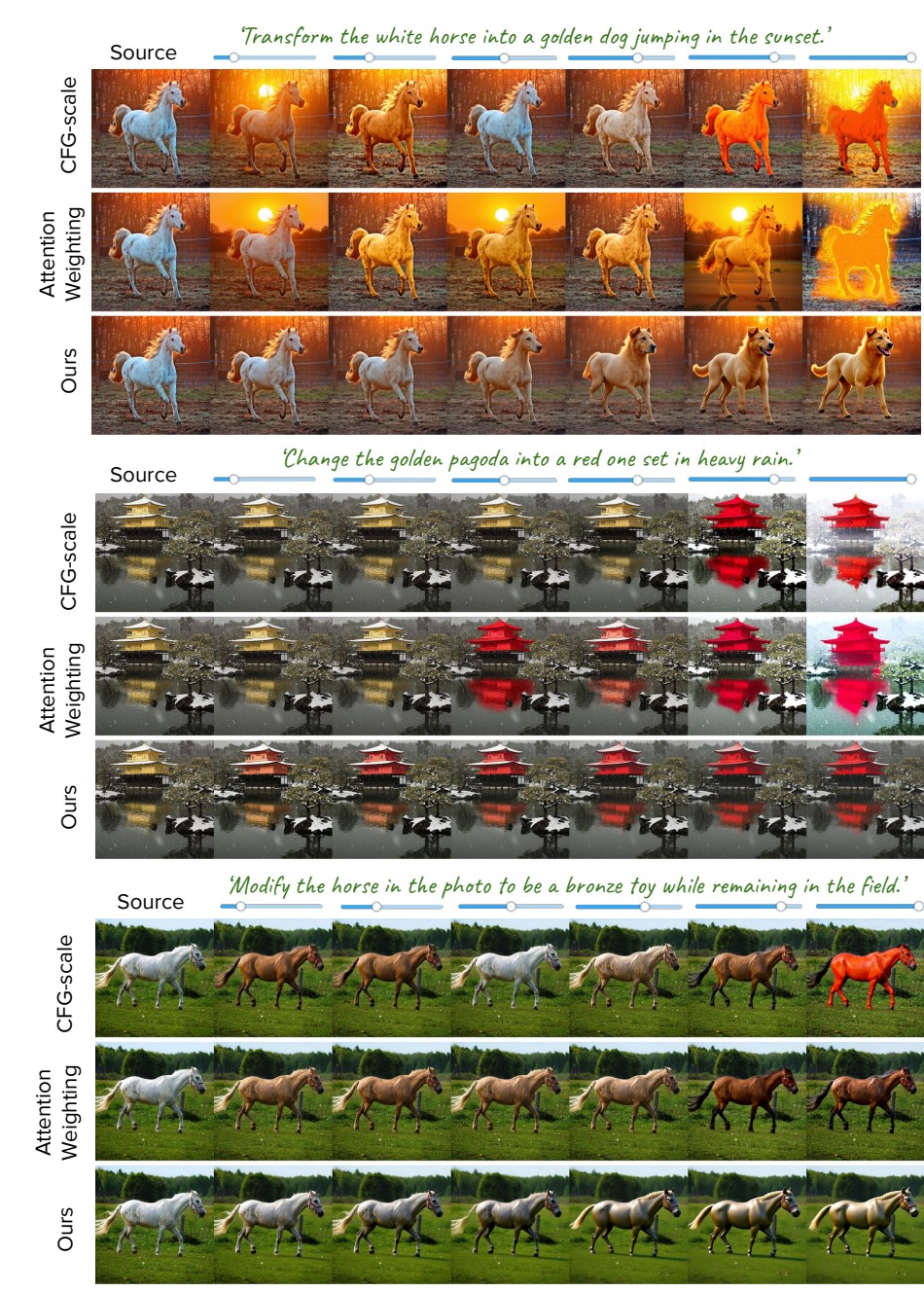

Figure 23: We compare with additional inference time baselines.

identity significantly. These abrupt transitions leads to a very high value for $\delta_{smooth}$ smoothness metric.

## A.9 FAILURE CASE - EXTRAPOLATION BEYOND THE TRAINING STRENGTH $s > 1$

One of the failure case of our method is in extrapolating edits beyond strength value $s = 1$. Our method either does not perform the edits for $s > 1$ or reduces the extent of the edit as shown in Fig. 24.

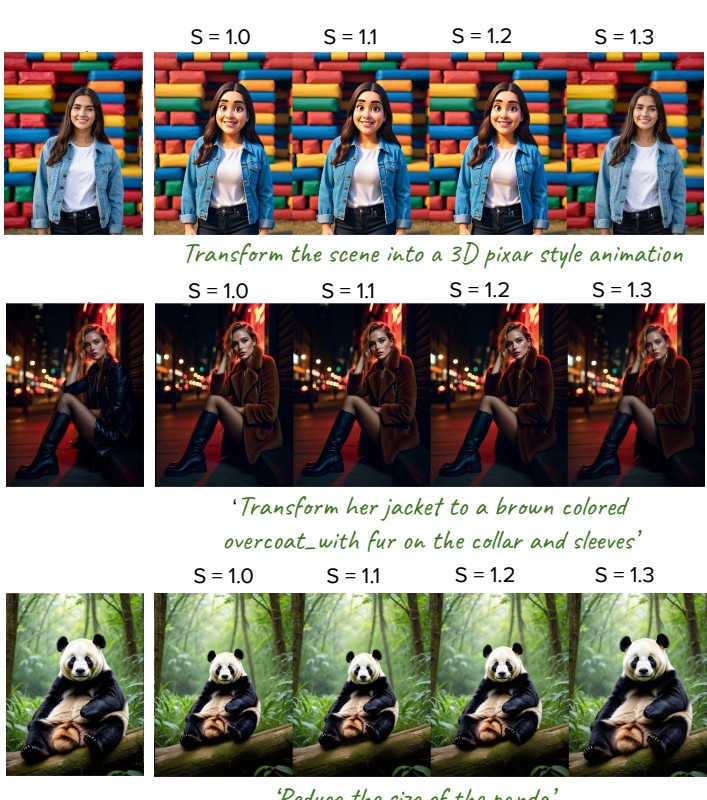

Figure 24: **Extrapolation of edit strengths**. One of the failure case of our method is it cannot generate edits with extrapolation well. In most cases, either it recreates the full edit image ($s = 1$), or reduce the extent of edit in extrapolation region.

## A.10 LLM USED IN WRITING THE PAPER

We have used LLM to do grammatical changes or rephrasing at a sentence level in the paper text. The authors of this paper are responsible for all the content of this paper.

