# OpenReview forum: "Kontinuous Kontext: Continuous Strength Control for Instruction-based Image Editing"
_ICLR.cc/2026/Conference — ICLR 2026 Conference Withdrawn Submission_

### Official Review · Reviewer_f8Be · 2025-10-29

**Soundness:** 2
**Presentation:** 2
**Contribution:** 2
**Rating:** 2
**Confidence:** 4

**Summary:**

This paper proposes Kontinuous Kontext, an instruction-driven image editing model that adds a scalar “edit strength” control to produce smooth trajectories from no edit to full edit. Concretely, the authors (1) synthesize training tuples *(source, instruction, strength, target)* by first generating a full-strength edit with Flux-Kontext and then producing intermediate-strength images via a diffusion morphing method (Freemorph), followed by filtering; and (2) learn a lightweight projector that maps *(strength, pooled text embedding)* to offsets in the modulation space of a DiT-based editor, thereby adjusting edit intensity. Evaluation reports smoother transitions and competitive instruction alignment on PIEBench (with add/remove removed) plus a small user study.

**Strengths:**

- Simple, engineering-friendly mechanism for continuous edit strength in an instruction-based editor (small projector, minimal overhead).
- Clear data pipeline and filtering criteria; reproducibility statement promises code/data release.
- Qualitative results show smooth trajectories on several categories.

**Weaknesses:**

- Strength supervision is entirely produced by Freemorph between a Flux-Kontext edit and the source; this may imprint particular morphing priors and penalize baselines differently. Include tests where strengths are annotated or verified on real edits, not generated morphs.
- Smoothness (δ_smooth) can be high even if edits undershoot semantics at intermediate s. Provide human preference vs. s and rank correlations (Spearman/Kendall) between CLIP-dir/δ_smooth and human judgments across categories.
- In domain-specific comparisons, CLIP-dir can be worse than specialized methods (e.g., material control vs. MARBLE), suggesting a trade-off that is not deeply analyzed.
- Dropping add/remove from PIEBench avoids a difficult (discrete) case; at least show failures/behavior there to contextualize applicability limits.
- Put identity preservation (e.g., face-ID/DINO features) and geometry accuracy front-and-center with per-strength metrics; current treatment is qualitative or buried in appendix.
- The ablation table focuses on projector inputs/placement and filtering; missing are scaling laws (data size vs. performance), projector capacity vs. overfitting, and sensitivity to text strength phrasing.

**Questions:**

1. How does instruction following vary with strength (e.g., CLIP-dir or human scores) per category and per s (not only aggregated)?
2. Can you report identity preservation metrics vs. s (and for people, a face-ID score), with CIs? Fail cases?
3. To de-bias from Freemorph, can you supply a real-image evaluation with human-rated intermediate strengths (or multiple independent morphers)?
4. What is the generation overhead (latency/VRAM) vs. Flux-Kontext across resolutions and batch sizes?
5. Please clarify the loss target in Eq. (1) and its connection to the base rectified-flow objective (any stability issues when conditioning on s?).

---

### Official Review · Reviewer_XiQc · 2025-10-31

**Soundness:** 2
**Presentation:** 3
**Contribution:** 2
**Rating:** 6
**Confidence:** 2

**Summary:**

The paper proposes a new complementary module for existing instruction-based image editing models, which allows for continous change and control of edit strength. Specifically, the paper proposes to learn a light-weight projector network that uses the strength scalar and editing instructions to predict the modulation coefficient for off-the-shelf flux-context model. The paper further proposes a pipeline to collect the corresponding training data, and present experiment results of the advanced smoothness and instruction-following results.

**Strengths:**

+ The paper proposes an interesting perspective regarding additional control for text-to-image generation.
+ The papper is well-organized and easy to follow.
+ The proposed idea in general is intuitive and effective.
+ Experiment results on PIEbench regarding transition smoothness and instruction-following quality seems decent, compared with the listed baselines. Human study further confirms the effectiveness of the proposed method.

**Weaknesses:**

- The proposed method seems heavily relying on the flux-kontext model. I kindly request the authors to comment on extending the method to other types of models/architectures.
- Quantitative results seem a bit limited to only PIEbench with \delta_smooth. Although perhaps not directly related, further discussions regarding compositional/multi-turn editing would strengthen claims.
- Human evaluation results are of relatively small scales.
- Admittely, I am not very familiar with the current progress of this particular sub-domain. I would therefore also like to hear other colleague reviewers' opinions.

**Questions:**

Please refer to the weaknesses section for detailed questions if any. Thanks.

---

### Official Review · Reviewer_XTD5 · 2025-10-31

**Soundness:** 3
**Presentation:** 3
**Contribution:** 3
**Rating:** 4
**Confidence:** 4

**Summary:**

To address the lack of fine-grained control over edit strength in instruction-based image editing, Kontinuous Kontext extends a state-of-the-art image editing model to accept an additional input, a scalar edit strength which is then paired with the edit instruction, enabling explicit control over the extent of the edit. It advances controllable image editing by bridging text instructions with fine-grained slider-based adjustments.

**Strengths:**

- The commendable quality of the paper is reflected in the rigor of its methodology and the extensive scope of its experiments. The data-synthesis pipeline interweaves LVLM-generated instructions, full-strength edits produced by Flux Kontext, and Freemorph interpolation, followed by a filtering stage that suppresses noise and elevates data fidelity. Quantitative evaluation on the standard PIEBench benchmark—including smoothness δ_smooth and CLIP-dir metrics—and a complementary user study consistently demonstrate superiority over both interpolation baselines and domain-specific competitors; the user study further corroborates the perceived superiority of the proposed approach.
- The paper is clearly written and logically structured. The method section incrementally elaborates the data-synthesis workflow (illustrated in Figure 3) and the model design (e.g., the projection network), whereas the experimental section unambiguously defines metrics and baselines. Key figures—such as the model architecture (Figure 6) and comparative results (Figure 9)—effectively facilitate comprehension. Despite the density of technical detail, the narrative remains coherent, enhancing reproducibility.
- The work manifests broad applicability and practical value. It introduces a new axis of fine-grained control, encompassing edits ranging from local attributes to global transformations. The paper further underscores the generality of the modulation space, suggesting significant potential impact on future research in controllable generation.

**Weaknesses:**

- The paper's data synthesis pipeline heavily relies on existing generative models, which introduces potential biases and errors. Although mitigated through filtering at the dataset level, the artificially set filtering thresholds may not completely eliminate all anomalies. The paper acknowledges insufficient support for discrete transformations (e.g., object insertion/removal) but does not delve deeply into alternative solutions.
- The primary contribution of this work is generating data based on existing models and training a model for continuous-strength image editing using this data with continuous strength annotations; however, the theoretical innovations involved are relatively limited.

**Questions:**

- Regarding innovation, the theoretical contributions of this paper, particularly in terms of the detail of the discussion, are somewhat insufficient. It is noted that the ablation experiments show significant impacts of various components on the experimental results. Could the underlying principles be elaborated to demonstrate that the model's effectiveness is not merely due to the extensiveness of the dataset? For instance, further explanation could be provided concerning the statement, "the modulation space of DiT models is highly disentangled and enables fine-grained control of attributes in text-to-image generation," and the specific threshold values set for "Poor Inversion" in the Data Filtering section.
- Continuous-strength image editing is time-sensitive in application, as users have low tolerance for waiting times after adjusting the edit strength (especially via a slider interface). The paper only provides a qualitative assessment of other baselines: "Further, these baselines are slower as they require a cascade of models for slider-based editing." Does the paper include specific experimental data on the model's generation time, particularly concerning changes only in edit strength?

---

### Official Review · Reviewer_aBqf · 2025-11-03

**Soundness:** 3
**Presentation:** 4
**Contribution:** 3
**Rating:** 8
**Confidence:** 4

**Summary:**

This paper introduces Kontinuous Kontext, a practical and well-engineered method for adding continuous edit strength control to instruction-based image editing models. The core idea is simple and effective: add a scalar “strength” input and map it (via a small MLP “projector”) together with a pooled text embedding into offsets in the model’s modulation space — i.e., yshift / yscale, so that strength manipulates the modulation parameters rather than raw tokens. The authors synthesize training tuples (source, instruction, strength, edited image) using a chain of generative components and then apply careful filtering to keep only smooth, well-behaved trajectories. The paper is well written and easy to understand. The result is a single unified model that provides smooth slider control across many edit categories (style, material, color, shape) without attribute-specific training.

**Strengths:**

1. Novel, pragmatic control mechanism — mapping scalar + pooled text embedding into modulation offsets (yshift/yscale) is simple, low-parameter, and directly targets the controllable latent (modulation) rather than forcing strength through token space where transitions are not smooth.
2. Strong focus on realistic training supervision — the authors share how to get strength-annotated tuples; they synthesize them using a chain of existing models and then apply explicit LPIPS / KL filtering to remove bad trajectories, which materially improves results (dataset reduced from ~110k → ~64.6k). This data curation is an important, reproducible contribution.
3. Good quantitative and qualitative results : Metrics aligned with human judgment — second-order triangle deficit (δ₂) computed with DreamSim correlates best with user preferences for smoothness (they ran a user study for this). Ablations show the importance of: (a) injecting the pooled text embedding into the projector, (b) injecting into modulation space rather than text embedding space, and (c) rigorous filtering of synthetic trajectories. These ablations support the core design choices.
4. Broad applicability across edit categories — unlike attribute-specific LoRA/adapters, the method works on stylization, materials, color, shapes and some geometric edits without per-attribute training, which is valuable for general editors.

**Weaknesses:**

1. Evaluation scope and baseline parity are limited in places: The main benchmark is PIEBench (540 images, add/remove removed). Domain-specific comparisons (ConceptSliders, MARBLE) are on small subsets (44 / 40 images respectively) or on generated images, which risks mismatched experimental setups and could lead to perceived cherry-picking. The paper should either expand these comparisons or clearly justify the mismatches.
2. Applicability of design choices to different architectures; The paper presents design choices specific to Flux Kontext, explaining how these changes will generalize to other architectures will help drive the conversation on Inherited limitations from the base model.
3. Limited extrapolation beyond trained range. The model struggles when asked to extrapolate to strengths beyond the training distribution (s>1) — it either saturates or collapses — which is a practical limitation for UI sliders that sometimes want extrapolation. An explanation on this will help and how the model generalizes to unseen concepts.

**Questions:**

1. How sensitive is the method to filtering thresholds (the KL > 0.15 number) and LPIPS cutoff choices? Can authors provide guidance for practitioners on how to tune filtering when using other morphing/inversion tools?
2. Can the authors quantify compute / latency / runtime for interactive slider usage compared to the interpolation baselines (DiffMorpher / Freemorph / WAN)? For real interactive tools, latency matters as much as smoothness ?
3. Can authors show Human-edit robustness test by evaluate on a subset of real human instructions (InstructPix2Pix test split or a small gathered set) and report δ₂, CLIP-dir, identity curves. This will test phrasing robustness.
4. Can authors compare to matched ConceptSliders/MARBLE runs: Run each method with the same input images and prompt templates on PIEBench categories where applicable (or alternatively run Kontinuous Kontext on their published datasets) and report per-category metrics

---

### Author Response · Authors · 2025-11-13
**Common response to all the reviewers**

Thanks for reviewing our paper and providing insightful feedback. Here we answer the common concerns raised by multiple reviewers.

**Inference time analysis**: Our method adds no significant overhead to Flux-Kontext inference. The lightweight 4-layer MLP projector introduces negligible latency and memory overhead.

*Interpolation baselines*: All the interpolation baselines (Freemorph, Diffmorpher, WAN), require one denoising pass of the Flux-Kontext model to generate the full edit. This is followed by one diffusion pass per edit strength of the image diffusion model for Freemorph and Diffmorpher and a single (but costlier) diffusion pass of the video diffusion model for WAN.

*Domain-specific baselines*: The domain-specific methods (MARBLE, ConceptSliders) and ours require a single forward diffusion pass per edit strength. Notably, unlike domain-specific methods (MARBLE, ConceptSliders), our approach requires no attribute-specific optimization for slider control.

**Discrete edits**: Recent instruction-based editing models support diverse edit types via natural text prompts. While text is well-suited to specify discrete edits, it lacks the ability to express continuous edits. For instance, ‘make him smile’ does not convey what should be the extent of the smile. Many real-world edits are inherently continuous and benefit from explicit control over edit intensity (e.g., degree of stylization or attribute/shape change). Our work focuses on such continuous edits, introducing precise strength control for instruction-based editing models. Continuous control for discrete edits (e.g., object addition or removal) is not meaningful; for example, “add 0.5 of a hat” lacks a clear semantic interpretation.

**Model architecture**: Our design for injecting the output of the slider projector into the modulation space of the DiT model is general. Since most recent instruction-based image editing models employ DiT blocks, our approach can, in principle, be readily adapted to other DiT-based architectures as well.

---

### Note · Authors · 2025-11-13

**Comment:**

I wish to **withdraw** the paper on behalf of myself all the co-authors.

**Withdrawal Confirmation:**

I have read and agree with the venue's withdrawal policy on behalf of myself and my co-authors.